# Visual Perception by Large Language Model's Weights

**Feipeng Ma**[1,2,*] **Hongwei Xue**[1,2,3,†] **Yizhou Zhou**[2,‡] **Guangting Wang**[2], **Fengyun Rao**[2]
**Shilin Yan**[4], **Yueyi Zhang**[1,‡] **Siying Wu**[5], **Mike Zheng Shou**[3], **Xiaoyan Sun**[1,5,‡]

[1]University of Science and Technology of China    [2]WeChat, Tencent Inc.
[3]Show Lab, National University of Singapore    [4]Fudan University
[5]Institute of Artificial Intelligence, Hefei Comprehensive National Science Center
{mafp,xuehongwei}@mail.ustc.edu.cn
harryizzhou@tencent.com, {zhyuey,sunxiaoyan}@ustc.edu.cn

## Abstract

Existing Multimodal Large Language Models (MLLMs) follow the paradigm that perceives visual information by aligning visual features with the input space of Large Language Models (LLMs) and concatenating visual tokens with text tokens to form a unified sequence input for LLMs. These methods demonstrate promising results on various vision-language tasks but are limited by the high computational effort due to the extended input sequence resulting from the involvement of visual tokens. In this paper, instead of input space alignment, we propose a novel parameter space alignment paradigm that represents visual information as model weights. For each input image, we use a vision encoder to extract visual features, convert features into perceptual weights, and merge the perceptual weights with LLM's weights. In this way, the input of LLM does not require visual tokens, which reduces the length of the input sequence and greatly improves efficiency. Following this paradigm, we propose VLoRA with the perceptual weights generator. The perceptual weights generator is designed to convert visual features to perceptual weights with low-rank property, exhibiting a form similar to LoRA. The experimental results show that our VLoRA achieves comparable performance on various benchmarks for MLLMs, while significantly reducing the computational costs for both training and inference. Code and models are released at https://github.com/FeipengMa6/VLoRA.

## 1 Introduction

Large language models (LLMs) [57, 65, 47] have achieved promising performance on most natural language tasks and have shown great generalization ability in solving real-world problems. Derived from LLMs, multimodal large language models (MLLMs) [36, 66, 4, 62, 55, 48] take a step toward artificial general intelligence (AGI) by perceiving visual information from the real world. Therefore, the way of perceiving visual information is the key to moving from LLM to MLLM.

To perceive visual information, recent MLLMs follow an input space alignment paradigm that aligns visual features with the input space of LLM and concatenates visual tokens with text tokens to form a unified sequence as input for LLM. For instance, LLaVA [36] uses CLIP-ViT-L-14 [50] as the visual encoder and introduces a linear projector to align the visual tokens with the input space of LLM. Monkey [31] divides input images into uniform patches and equips individual adapters for each

---

*This work was performed while Feipeng Ma and Hongwei Xue were interns at WeChat, Tencent Inc.
†Project Leader.
‡Corresponding authors.

patch to handle high-resolution images. Recent work [56] also identifies the visual shortcomings of CLIP for MLLMs as "CLIP-blind pairs" and integrates vision self-supervised learning features with MLLM to address this issue. DeepSeek-VL [41] and Sphinx [32] also adopt hybrid vision encoders. Vary [58] identifies that a fixed vision vocabulary limits the dense and fine-grained visual perception and introduces a new vocabulary to address this issue.

Despite these efforts to advance MLLM in visual perception, the paradigm of input space alignment remains unchanged, which can result in computational inefficiency for both training and inference. The computational cost of MLLM is concentrated on the attention mechanism of LLM, which is $O(n^2)$ when the length of the input sequence is $n$. Using ViT-L-14 as the vision encoder, a 224×224 low-resolution image can result in 256 visual tokens, and the length increases to 576 when the resolution slightly raises to 336×336. Considering high-resolution images, some works [32, 35, 31, 11] split an image into multiple sub-images for capturing fine-grained information, leading to a significantly higher number of visual tokens. For instance, Sphinx-2k [32] adopts 2,890 visual tokens, while InternLM-Xcomposer2-4KHD [11] even uses up to 8,737 visual tokens. Concatenating such a long sequence of visual tokens to text tokens results in a dramatic increase in computational overhead for both training and inference. Specifically, current MLLMs are usually pre-trained on web-crawled image-text pairs, which usually have very short texts, with an average word count of 10.95 for LAION-2B [51] and 8.99 for LAION-COCO [1]. As a result, the number of visual tokens during the pre-training stage is about 20 to 50 times the number of text tokens, which suggests that the involvement of visual tokens seriously affects the efficiency of the pre-training. Some works [27, 9, 24] employ resamplers to reduce the number of visual tokens to a fixed count but still follow the input space alignment paradigm and introduce extra visual tokens for LLMs.

To address this issue, we explore a novel parameter space alignment paradigm where visual information is represented as LLM's weights. As shown in Fig. 1, for an input image, we use a vision encoder to extract visual features. Then, the visual features are converted to perceptual weights, which represent visual information as model weights. The perceptual weights can be directly merged with LLM's weights. Thus, the visual information is merged into LLM in the form of weights, eliminating the need for visual tokens in the LLM's input and significantly improving efficiency. Building on this paradigm, we introduce VLoRA, which contains the perceptual weights generator. The perceptual weight generator is designed to convert visual features to perceptual weights. LLMs usually contain a large number of parameters, for feasibility and efficiency, perceptual weights are designed with a low-rank property. Thus the generated perceptual weights are similar to the form of LoRA weights.

Our contributions are summarised as follows:

1. We explore a novel paradigm for MLLMs that aligns visual features with the parameter space of LLMs, which highly improves the efficiency of MLLMs

2. Based on this paradigm, we propose VLoRA and design the perceptual weights generator that generates low-rank perceptual weights.

3. Experimental results demonstrate the effectiveness and efficiency of our approach. We obtain results comparable to those of state-of-the-art MLLMs on various benchmarks, including MMBench, ScienceQA, HallusionBench, and MMMU.

## 2 Related Works

**Multimodal Large Language Models.** Current MLLMs are developed from LLMs by aligning visual features into the input space of LLMs. Many efforts have been made to explore introducing visual perception capability for LLMs. LLaVA [36] connects the visual encoder of CLIP to the Vicuna [65] with a linear projector. Further research that follows this paradigm focuses on improving MLLMs from the perspective of vision encoder and projector DeepSeek-VL [41] use SigLip [61] to extract high-level semantic features and use SAM-B [22] to process low-level features. Tong *et al.* [56] finds that visually distinct images can be encoded as similar due to the shortcoming of CLIP and integrates vision self-supervised learning features with CLIP features. Sphinx [32] ensembles various vision backbones that have different architectures, pre-training paradigms, and information granularities. These works input the entire visual tokens sequence into the LLM, which can lead to a high computational cost during training and inference. Specifically, LLaVA [34] and DeepSeek-VL [41] utilize 576 visual tokens, Sphinx-2k [32] employs 2,890 visual tokens, and InternLM-XComposer2-4KHD [11] uses up to 8,737 tokens. Some works consider adopting cross-attention

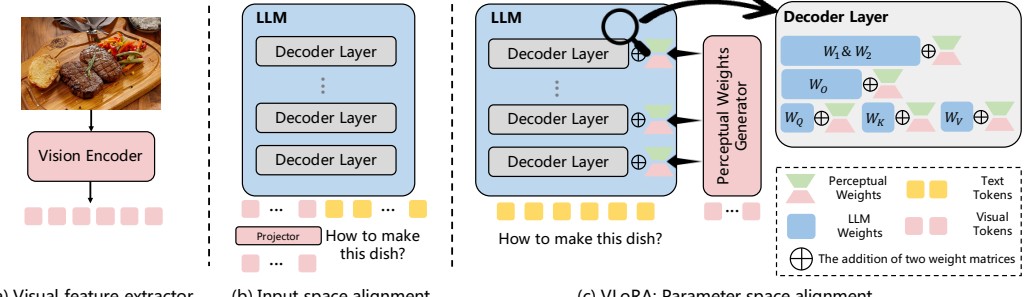

(a) Visual feature extractor  (b) Input space alignment  (c) VLoRA: Parameter space alignment

Figure 1: **Overview of the input space alignment and the parameter space alignment paradigms.** The input space alignment paradigm is aligning visual features with the input space of LLM and concatenating visual tokens with text tokens as input for LLM. Our proposed VLoRA follows the parameter space alignment paradigm that aligns visual features with the parameters of LLM and merges perceptual weights generated by the perceptual weights generator with LLM's weights.

architecture as the projector to improve efficiency. MiniGPT4-v1 [66] and BLIP series [27, 9] adopt Q-Former as the projector, which reduces the length of visual tokens to a fixed number of 64. Qwen-VL [5] uses a single-layer cross-attention module incorporated with 2D absolute positional encodings to avoid the potential loss of positional details. However, these improvements still follow the paradigm of aligning visual features to the input space of LLM, introducing extra computational overhead on LLM inference. Different from previous work, our VLoRA aligns visual features with the parameter space of LLM. The visual information can be represented as perceptual weights in LoRA format and merged into LLM's weights during inference.

**Parameter-Efficient Fine-Tuning.** Parameter-efficient fine-tuning (PEFT) is a key technique for fine-tuning large pre-trained models, including LLMs and MLLMs. PEFT methods freeze the backbone and only fine-tune a small number of parameters, which can be typically categorized into three classes: adapters [17, 49, 54, 63], prefix-tuning [29, 26, 38], and Low-Rank Adaption (LoRA) [18, 37, 10]. Houlsby *et al.* [17] design bottleneck adapters and insert two adapters into the transformer layers, one after the attention module and one after the feed-forward network. LLaMA-Adapter [63] inserts learnable prompts into $L$ of $N$ decoder layers and uses zero-initialized attention for stable training. Prefix-tuning [29] prepends a set of learnable prefix vectors at the query and key of the self-attention module for every layer. Prompt-tuning proposes to only prepend learnable vectors to the input prompt with no intermediate-layer prefixes. LoRA [18] uses learnable low-rank matrices to approximate the backbone's weight updates, and the low-rank matrices can be merged with the backbone during inference without extra inference burden. Considering the pre-training stage, current MLLMs usually freeze the unimodal backbones and project visual tokens through a learnable projector, then prepend visual tokens into the input sequence of LLMs, which can be seen as prefix-tuning methods. Our VLoRA is closer to the style of LoRA. Specifically, VLoRA generates low-rank perceptual weights, which can be seen as a generated visual parameters matrix $\Delta W_A \in \mathbb{R}^{h \times r}$ multiplied with a learnable matrix $\Delta W_B \in \mathbb{R}^{r \times h}$. Similar to LoRA, the perceptual weights can be injected into LLMs' weights without introducing extra inference overhead.

**HyperNetwork.** HyperNetwork is a technique that employs one network to generate the weights for another network. HyperNetworks [16] proposes static hypternetwork for CNN and dynamic hypternetwork for RNN. HyperFormer [43] proposes hypterformer to generate adapter parameters for all layers and multiple tasks using shared hypternetworks. The parameter generation of both methods is designed on task-level for pre-defined tasks. HyperPELT [64] employs a shared hypernetwork that generates weights for prefix-tuning and adapter-tuning modules. MemVP [20] concatenates visual prompts with FFN weights to inject visual knowledge. In contrast to HyperNetworks and HyperFormer, 1) VLoRA focuses on sample-level parameter generation, the generated LoRA weights are conditioned on the input image without pre-defining tasks during training. Since the goal of MLLMs is to address a wide range of tasks and problems that are difficult to fully define in advance, task-level adaptation is unsuitable for MLLMs. 2) VLoRA utilizes the generated parameters in LoRA way. Sample-level parameter generation can lead to significant changes in model parameters.

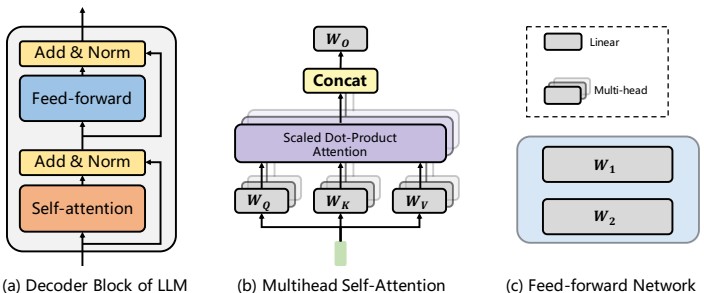

(a) Decoder Block of LLM    (b) Multihead Self-Attention    (c) Feed-forward Network

Figure 2: **Details of the LLM Decoder Block**. (a) illustrates the details of the LLM decoder block, including the multi-head self-attention module and the feed-forward network. (b) provides a detailed view of the multi-head self-attention module, which incorporates four types of weights: $W_Q$, $W_K$, $W_V$, and $W_O$. (c) depicts the feed-forward network, which consists of the weights $W_1$ and $W_2$.

VLoRA, adopting the LoRA method, can better maintain the inherent capability of the pre-trained LLM. Unlike HyperPELT and MemVP, 1) VLoRA can inject visual information at any linear module, offering flexibility. 2) Unlike task-level PEFT methods, VLoRA is sample-level, generating weights for individual input images. Our evaluations, mainly in zero-shot settings, demonstrate VLoRA's strong generalization ability.

## 3 Method

### 3.1 Preliminaries

In this subsection, we review the details of the decoder block in the current LLM. As shown in Fig. 2, the decoder block of LLM contains a self-attention module and a feed-forward network.

**Self-attention.** As shown in Fig. 2 (b), the self-attention module contains four types of linear layers: query $W_Q \in \mathbb{R}^{h \times d}$, key $W_K \in \mathbb{R}^{h \times d}$, value $W_V \in \mathbb{R}^{h \times d}$, and output $W_O \in \mathbb{R}^{h \times h}$. Here, $h$ represents the dimension of the hidden states of LLM, and $d$ represents the dimension of each attention head. For each input token $x_i \in \mathbb{R}^h$ in the input sequence $X = (x_1, x_2, ..., x_N)$, it is multiplied by linear layers $W_Q, W_K, W_V$, obtaining $X^q = X W_Q$, $X^k = X W_K$ and $X^v = X W_V$. Then, the attention operation is executed along the sequence dimension as follows:

$$\text{Attention}(X^q, X^k, X^v) = \text{softmax}(\frac{X^q X^{k^T}}{\sqrt{d}}) X^v. \tag{1}$$

The self-attention mechanism is performed on each head, and the outputs from different heads are concatenated and multiplied by output linear layer with weights $W_O$.

**Feed-forward Network.** As shown in Fig. 2 (c), the feed-forward network is an MLP with two fully connected layers and a non-linear activation function. The formulation can be written as follows:

$$\text{FFN}(x_i) = \phi(x_i W_1) W_2, \tag{2}$$

where $x_i$ is the input token, $\phi$ is the activation function, and $W_1$ and $W_2$ are the weights of two fully connected layers. To summarize, the decoder block of LLM has five types of weights, including $W_Q$, $W_K, W_V, W_O$ from the self-attention module, and $W_1, W_2$ from the feed-forward network.

### 3.2 Visual Perception by LLM's Weights

Previous MLLMs follow the paradigm of aligning the visual features with the input space of LLM and require additional visual tokens as LLM's input, which can lead to computational inefficiency. This inefficiency becomes more pronounced when encountering high-resolution or multiple images as the number of tokens increases drastically. To address this issue, we propose to align visual features with LLM's parameter space without introducing extra tokens into LLM's input.

To achieve this goal, we represent the visual information of the input image as perceptual weights and integrate them into the weights of LLM. This approach allows LLM to perceive visual information

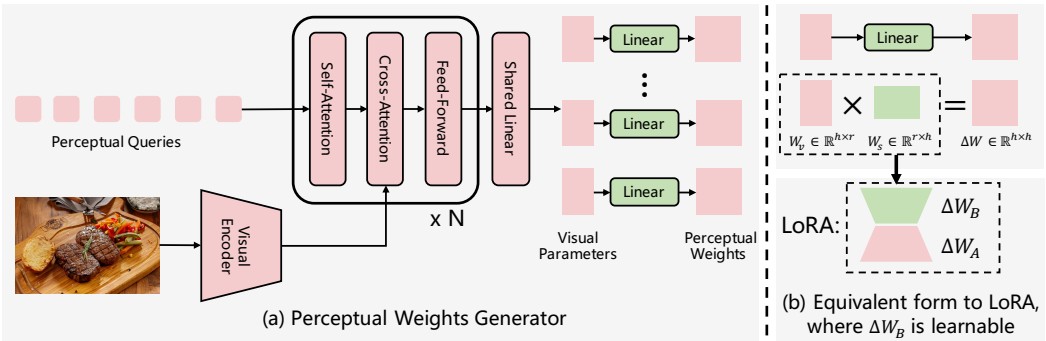

Figure 3: **Perceptual Weights Generator.** Figure (a) illustrates the pipeline of our perceptual weights generator. We set $k$ learnable perceptual queries, which interact with image features in $N$ decoder blocks, and obtain $k$ visual parameters. Then, a shared linear layer and $k$ independent linear layers are used to convert these visual parameters to perceptual weights $\Delta W$. Figure (b) demonstrates that our approach is formally consistent with LoRA.

without introducing extra tokens into the input. As mentioned in Sect. 3.1, LLM's decoder blocks have five types of weights. We use $W \in \mathbb{R}^{h \times h}$ to denote the weight matrix of LLM. For an input image $I$, we first adopt a vision encoder $f(\cdot)$ to extract the visual features $z = f(I)$, where $z \in \mathbb{R}^{c \times d_v}$, $c$ is the number of visual tokens, and $d_v$ is the dimension of visual features. Then, we design a perceptual weights generator $g(\cdot)$ to convert the visual features to perceptual weights $\Delta W \in \mathbb{R}^{h \times h}$. It is worth noting that, given that we want LLM to perceive visual information while preserving its language capabilities, $\Delta W$ is a low-rank matrix, which also helps to reduce the computation cost of the perceptual weights generator. With the generated perceptual weights $\Delta W$, we can directly merge it into the LLM's weights as:

$$\hat{W} = W + \Delta W. \tag{3}$$

By integrating the weights transferred from the visual features into the LLM's weights, the visual perception ability is naturally equipped. After merging the weights, no extra inference burden will be introduced for LLM. For any weights in each decoder block of LLM, we can generate the corresponding perceptual weights and integrate them into LLM's weights.

### 3.3 Perceptual Weights Generator

To convert visual features to perceptual weights $\Delta W \in \mathbb{R}^{h \times h}$, we propose the perceptual weights generator. Since each layer and each type of weight in LLM focus on different visual information, our perceptual weights generator needs to be able to generate weights corresponding to each of the LLM weights flexibly.

Inspired by DETR [6] and BLIP-2 [27], we design the perceptual weights generator as a decoder-only architecture with cross-attention layers to generate $\Delta W \in \mathbb{R}^{h \times h}$. As shown in Fig. 3 (a), the perceptual weights generator contains $N$ blocks, each comprising a self-attention module, a cross-attention module, and a feed-forward network. The hidden states dimension of the perceptual weights generator is $h_p$, where $h_p \ll h \cdot h$. We set $k$ learnable perceptual quires corresponding to the number of decoder blocks where we want to insert perceptual weights. For each block, the perceptual queries first pass through the self-attention module, then interact with visual features in the cross-attention module, and finally go through a feed-forward network. After $N$ blocks, we obtain $k$ features $p_v \in \mathbb{R}^{h_p}$. The features $p_v$ should be mapped to the target shape of perceptual weights $\Delta W \in \mathbb{R}^{h \times h}$. However, due to $h_p \ll h \cdot h$, directly mapping the dimensions of the $p_v$ from $h_p$ to $h \cdot h$ with a linear layer can introduce a large number of parameters, dramatically reducing the feasibility. Therefore, we consider introducing the low-rank property in this process. We adopt a shared linear layer $W_{share} \in \mathbb{R}^{h_p \times h \cdot r}$ to map all features $p_v$ from $h_p$ to $h \cdot r$ as follows:

$$W_v = p_v W_{share}, \tag{4}$$

where $r$ is the rank for perceptual weights and $W_v \in \mathbb{R}^{h \cdot r}$ is visual parameter.

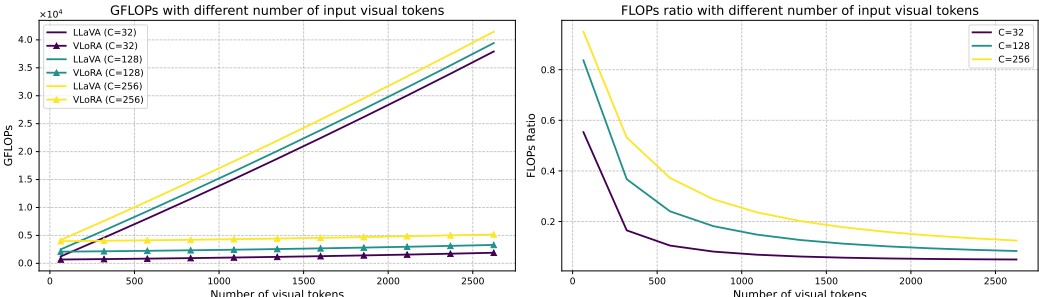

Figure 4: **Comparison of FLOPs.** This figure shows the FLOPs of LLaVA and VLoRA with different numbers of input visual tokens. The left subplot illustrates the change in GFLOPs, the right subplot plots the ratio of GFLOPs for VLoRA to LLaVA, and C denotes the number of text tokens.

And we reshape the output $W_v$ as $h \times r$. When ascending to the target dimension $h \times h$, $k$ independent linear layers $W_s \in \mathbb{R}^{r \times h}$ are used for each visual parameter and obtain $k$ perceptual weights $\Delta W$, this process can be formulated as follows:

$$\Delta W = W_v W_s. \tag{5}$$

Substituting Eq. (5) into Eq. (3), we get:

$$\hat{W} = W + \Delta W = W + W_v W_s. \tag{6}$$

Considering the low-rank property of $W_v$ and $W_s$, we can observe that Eq. (6) and LoRA [18] are of the same form, where $W_v$ corresponds to $\Delta W_A$ and $W_s$ corresponds to $\Delta W_B$. As illustrated in Fig. 3 (b), our perceptual weights generator can be seen as "LoRA weights generator" from the perspective of LoRA. This is because it generates $\Delta W_A$ and $\Delta W_B$ for weights of LLM. Our perceptual weights generator generates one type of perceptual weights for $k$ decoder blocks at a time. For generating multiple types of weights, we employ multiple perceptual weights generators.

### 3.4 Analysis of the Computational Cost

By not introducing additional visual tokens in the input of the LLM, our VLoRA achieves higher computational efficiency for both training and inference. We only consider the computational cost of LLM, as the computational overhead of our perceptual weights generator is negligible in comparison. We assume the LLM has $d$ blocks and hidden states dimension of $h$, the input text length is $C$, and the number of visual tokens is $L$. For convenience, we only consider the computational cost of the self-attention module and feed-forward network in LLM. The FLOPs of the self-attention module and the feed-forward network are $8Lh^2 + 4L^2h$ and $16Lh^2$. For previous MLLMs that align visual features to the input space of LLM, the FLOPs of LLM are $24(L + C)dh^2 + 4(L + C)^2dh$. For our VLoRA, the extra computational cost occurs in Eq. (6), where $\Delta W_A$ is multiplied with $\Delta W_B$. Assuming that we generate perceptual weights for all 5 types of weighs in $k$ decoder blocks. During training, we do not merge the perceptual weights with the LLM weights but use them as branches of the LLM weights. Therefore, the FLOPs are $24Cdh^2 + 4C^2dh + 24krh^2 + 12Ckh^2 + 14Ckh$. For inference, the perceptual weights can be merged into the LLM, and the FLOPs are $24Cdh^2 + 4C^2dh + 24krh^2 + 12kh^2$. Details of the FLOPs calculation are in the Appendix A. There is a small increase in the overhead of training compared to inference, and we compare by the training FLOPs. In Fig. 4, we compare the FLOPs of LLaVA and VLoRA. Our approach does not introduce additional computation as the number of visual tokens increases, and our FLOPs are only **8%** of LLaVA-v1.5's when the text length is 32.

## 4 Experiments

### 4.1 Implementation Details

**Model Settings.** We use Vicuna-7b-v1.5 [65] as our foundational LLM and CLIP-ViT-L-14 [50] as vision encoder. The perceptual weights generator is initialized randomly. For the perceptual weights

generator, we set the hidden size $h_p$ as 512, and the number of blocks $N$ as 8. The rank $r$ of perceptual weights is 64. The number of perceptual queries is 8, which means that we insert perceptual weights $\Delta W$ only on 8 blocks, and in the implementation, for Vicuna-7b-v1.5 with 32 blocks, we insert $\Delta W$ every 4 blocks. For better visual perceptual ability, we insert $\Delta W$ for all five types of weights in LLM. It is worth noting that the last $k$ linear layers of the perceptual weights generator are zero-initialized as they are equivalent to the $\Delta W_B$ of LoRA weights, which are initialized as zero for training stability.

**Pre-training Data.** During pre-training, we use image-text pairs to train our model. Specifically, we use a subset of CapsFusion-120M [59] with 30 million image-text pairs. CapsFusion-120M randomly collects image-text pairs from LAION-COCO [1], which contains both web-crawled and synthetic captions generated by BLIP [28]. Then, a fine-tuned LLM is used to integrate both types of captions.

**Pre-training Configuration.** We freeze the weights of LLM and visual encoder in the pre-training stage, making only the perceptual weights generator trainable. We use the AdamW [40] optimizer with a learning rate of $5e$-5, which follows a linear warm-up and then a cosine decay schedule. The pre-training is conducted with a total batch size of 768 for 40,000 iterations. The input images are resized to a resolution of $336 \times 336$. The pre-training stage uses 24 NVIDIA H800 GPUs for 7 hours.

**Fine-tuning Data.** For supervised fine-tuning, we adopt the same data as LLaVA-v1.5. Specifically, the supervised fine-tuning data is constructed with VQAv2 [13], GQA [19], OKVQA [45], OCRVQA [46], A-OKVQA [52], TextCaps [53], RefCOCO [44, 21], Visual Genome [23], ShareGPT [2], and LLaVA-Insturct [36], with a total of 665K conversation data.

**Fine-tuning Configuration.** During the fine-tuning stage, we freeze the vision encoder and update the weights of the perceptual weights generator and LLM. The learning rate is set to $5e$-5 and the learning rate schedule is the same as in the pre-training stage. The global batch size is 128. We train for one epoch on 8 NVIDIA H800 GPUs, which takes 2 hours.

### 4.2 Benchmarks for Evaluation

**MMBench & CCBench.** MMBench [39] is a comprehensive multimodal benchmark designed to evaluate the performance of MLLMs. It includes over 3,000 multiple-choice questions covering 20 ability categories. The evaluation is divided into perceptual and reasoning dimensions and subdivided into 20 categories. CCBench [39], released by the MMBench team, is designed for evaluating MLLMs in the domain of Chinese Culture.

**MME.** MME [12] also measures the advanced MLLMs in terms of perception and cognition, with a total of 14 subtasks. To minimize the influence of prompt engineering on MLLMs, the instructions of MME are designed as simple binary responses: "please answer yes or no".

**ScienceQA.** ScienceQA [42] is constructed from elementary and high school science curricula. Questions of ScienceQA span three subjects: natural science, language science, and social science. We use samples with images from the validation set to evaluate MLLMs.

**HallusionBench.** HallusionBench [14] is designed for evaluating image-context reasoning, including 346 images paired with 1129 questions crafted by human experts. Unlike other benchmarks [15, 30, 33] that focus on object hallucinations with limited topics and visual input types, HallusionBench considers both language hallucinations and visual illusions across a diverse range of topics.

**MMMU.** MMMU [60] collects 11.5K multimodal questions from college exams, quizzes, and textbooks, covering six core disciplines, spanning 30 subjects and 183 subfields, and comprising 30 heterogeneous image types. MMMU is more challenging than existing benchmarks due to the demand for college-level domain-specific knowledge.

### 4.3 Comparison with State-of-the-arts

Tab. 1 compares our VLoRA with other state-of-the-art MLLMs on six MLLM benchmarks. The results are obtained from OpenCompass [8]. Unlike other MLLMs, our VLoRA does not require any visual tokens during LLM inference and has only **8%** of the computational overhead of LLaVA-v1.5 when the text length is 32. On most benchmarks, VLoRA outperforms InstructBLIP, MiniGPT-4, Idefics-instruct, and OpenFlamingo v2. Compared with Qwen-VL-Chat pre-trained on 1.4B image-text pairs, VLoRA has a higher score of 3.7 on MMBench and 1.3 on ScienceQA. Compared with LLaVA-v1.5, VLoRA can achieve comparable performance on MMBench, ScienceQA, and

Table 1: Comparisons on six MLLM benchmarks, including MMBench, MME, ScienceQA, HallusionBench, MMMU, and CCBench. *vis. tok.* denotes the number of visual tokens involved in the LLM. Bolded numbers indicate the best results, and underlined numbers are the second-best results.

| Model | Size | # *vis. tok.* | MMBench | MME | ScienceQA | HallusionBench | MMMU | CCBench |
|---|---|---|---|---|---|---|---|---|
| InstructBLIP [9] | 8B | 32 | 36.0 | 1137.1 | 54.7 | 31.2 | 30.6 | 12.7 |
| MiniGPT-4-v1 [66] | 7B | 32 | 12.2 | 770.6 | 39.0 | 31.9 | 23.6 | 1.8 |
| MiniGPT-4-v2 [7] | 7B | 256 | 24.3 | 708.4 | 54.1 | 30.0 | 25.0 | 1.4 |
| Idefics-instruct [25] | 9B | 64 | 48.2 | 942 | 51.6 | 27.3 | 18.4 | 7.8 |
| OpenFlamingo v2 [3, 4] | 9B | 64 | 6.6 | 535 | 45.7 | 29.4 | 28.2 | 6.3 |
| Qwen-VL [5] | 9.6B | 256 | 38.2 | 334.1 | 57.7 | 29.9 | 29.6 | 6.1 |
| Qwen-VL-Chat [5] | 9.6B | 256 | 60.6 | 1467.8 | 65.5 | **36.8** | **37.0** | **41.2** |
| LLaVA-v1.5 [34] | 7.2B | 576 | **64.3** | **1510.7** | **66.8** | 27.6 | 35.7 | 27.5 |
| VLoRA | 7.8B | **0** | 63.4 | 1311.3 | 66.4 | 26.4 | 33.7 | 28.6 |

Table 2: Comparison to LLaVA-v1.5 with various settings on six MLLM benchmarks, including MMBench, MME, ScienceQA, HallusionBench, MMMU, and CCBench. PT data represents the pre-training data. *vis. tok.* denotes the number of visual tokens involved in LLM.

| Model | PT data | # vis. tok. | MMBench | MME | ScienceQA | HallusionBench | MMMU | CCBench |
|---|---|---|---|---|---|---|---|---|
| LLaVA-7b-v1.5 | blip-558k | 576 | 64.3 | 1510.7 | 66.8 | 27.6 | 35.7 | 27.5 |
| LLaVA-7b-v1.5 | CapsFus-30m | 576 | 64.6 | 1470.0 | 67.7 | 27.4 | 33.8 | 25.3 |
| LLaVA-7b-v1.5-QFormer | CapsFus-30m | 128 | 60.7 | 1241.5 | 67.3 | 26.7 | 33.8 | 25.3 |
| VLoRA | CapsFus-30m | **0** | 63.4 | 1311.3 | 66.4 | 26.4 | 33.7 | 28.6 |

HallusionBench and even better performance on CCBench. However, the results on MME fall short of LLaVA-v1.5 since our perceptual weights generator is randomly initialized and necessitates more image-text pair data during the pre-training stage. To verify this, in Tab. 2, we reproduce LLaVA-v1.5 by replacing the projector with a randomly initialized Q-Former and achieve similar results on MME. Our VLoRA achieves comparable performance to state-of-the-art MLLMs without introducing visual tokens as LLM inputs, drastically reducing computational overhead.

## 5 Ablation Study

Currently, the performance of MLLMs is significantly affected by the foundational LLMs and the training data, including pre-training data and supervised fine-tuning data. To explore the effectiveness of our proposed paradigm and model, we perform a fair comparison with LLaVA-v1.5 [36] by adopting the same foundation LLM and training data in this section. Then, with this setting, we also explore the impact of different settings of each component on performance.

### 5.1 Comparison with LLaVA-v1.5

To ensure a fair comparison with LLaVA-v1.5, we reproduce LLaVA-v1.5 with the same setting as our VLoRA, including the pre-training and supervised fine-tuning data. Furthermore, to eliminate the influence of the difference in the projector, we replace the project of LLaVA-v1.5 as a randomly initialized Q-Former, which has the same number of blocks and hidden size as our perceptual weights generator. The training is conducted using the same pre-training and fine-tuning data as VLoRA.

In Tab. 2, the second row is the results of LLaVA-v1.5 pre-training on CapsFus-30m. With more pre-training data, LLaVA-v1.5 doesn't achieve significant improvement on MLLM benchmarks but rather a drop on MME, HallusionBench, MMMU, and CCBench. Our VLoRA is still comparable with the LLaVA-v1.5 training on the same data. The third row is the results of LLaVA-v1.5 with Q-Former, which is pre-trained on CapsFus-30m. We set the number of learnable queries as 128, thus the number of visual tokens is 128. Except for being slightly lower in ScienceQA and HallusionBench, our VLoRA is significantly better on other MLLM benchmarks. These results demonstrate that our approach is comparable to or even better than LLaVA-v1.5 with consistent settings.

Table 3: The impact of weights type that equipped perceptual weights. q, k, v, and o denote the query, key, value, and output weights in the self-attention module, respectively. m denotes the weights of the feed-forward network.

| Weights type | MMBench | MME | ScienceQA | HallusionBench | MMMU | CCBench |
|---|---|---|---|---|---|---|
| qkvom | 63.4 | 1311.3 | 66.4 | 26.4 | 33.7 | 28.6 |
| qkvm | 59.6 | 1227.5 | 64.6 | 23.4 | 33.2 | 24.9 |
| qkv | 59.4 | 1267.9 | 65.8 | 23.2 | 33.9 | 28.8 |
| qko | 57.2 | 1240.5 | 64.0 | 23.4 | 34.6 | 24.9 |
| qk | 53.3 | 1169.8 | 65.0 | 23.5 | 32.0 | 21.8 |

Table 4: The impact of perceptual weights' rank. The rank of the generated perceptual weights indicates the extent of visual information compression.

| Rank | MMBench | MME | ScienceQA | HallusionBench | MMMU | CCBench |
|---|---|---|---|---|---|---|
| $r = 16$ | 59.4 | 1212.7 | 67.1 | 22.9 | 33.7 | 24.5 |
| $r = 32$ | 60.7 | 1235.6 | 67.2 | 23.5 | 33.2 | 25.3 |
| $r = 64$ | 63.4 | 1311.3 | 66.4 | 26.4 | 33.7 | 28.6 |
| $r = 128$ | 61.0 | 1228.4 | 68.0 | 23.8 | 33.4 | 26.7 |

## 5.2 Analysis of each component

To further analyze VLoRA, we explore the impact of each component, including the type of weights that equipped perceptual weights, the rank of perceptual weights, and the number of blocks of perceptual weights generator.

**The type of weights that equipped perceptual weights.** As we mentioned in Sect. 3.1, there are five types of weights in the decoder block of LLM, which are **q**uery, **k**ey, **v**alue, **o**utput, and **m**lp. We explore the impact of inserting perceptual weights for different types of LLM weights. As shown in Tab. 3, we compare different combinations, including qkvom, qkvm, qkv, qko, and qk. The model that equipped perceptual weights for all types of weights can achieve the best performance on most benchmarks. We notice that the performance of qkv is much better than qk. This suggests that the value matrix is essential for visual perception since the output of the value matrix will be weighted and summed, involving the results of the self-attention module.

**The rank of perceptual weights.** The rank of the generated perceptual weights represents the degree of visual information compression. The smaller the rank, the more compressed the visual information. We compare the performance of rank $r$ from 16 to 128 in Tab. 4. When the $r = 16$, the visual information is compressed severely in perceptual weights. However, LLM with such low-rank perceptual weights can still perceive visual information. From $r = 16$ to $r = 64$, the performance on MMBench, MME, HallusionBench, and CCBench improves with increasing rank. Specifically, the score of MMBench increases from 57.6 to 63.4, and the score of MME increases from 1163.8 to 1311.3. When the rank reaches 128, VLoRA's performance declines across these benchmarks. The reason might be that the visual information becomes redundant, and a large rank may introduce noise into the perceptual weights, which hurts LLM's capability.

**The number of blocks of perceptual weights generator.** To explore the influence of the perceptual weights generator, we perform experiments with different numbers of blocks in the perceptual weights generator. In Tab. 5, we observe that the performance of the weights generator with 8 blocks is better than with 4 blocks. However, when it comes to $N = 12$, the scores on ScienceQA and CCBench are higher than with 8 blocks, but performance drops on other benchmarks. This suggests that while a stronger perceptual weights generator can achieve better performance, there is no benefit to increasing the number of blocks after the threshold is reached.

## 6 Conclusion

In this paper, instead of aligning visual features with the input space of LLM, we propose VLoRA to align visual features with the parameter space of LLM. By not introducing visual tokens into

Table 5: The impact of different numbers of blocks of perceptual weights generator.

| Blocks | MMBench | MME | ScienceQA | HallusionBench | MMMU | CCBench |
|--------|---------|--------|-----------|----------------|------|---------|
| $N = 4$ | 60.7 | 1289.3 | 63.9 | 24.4 | 32.0 | 26.7 |
| $N = 8$ | 63.4 | 1311.3 | 66.4 | 26.4 | 33.7 | 28.6 |
| $N = 12$ | 61.3 | 1289.3 | 67.1 | 25.5 | 33.8 | 30.2 |

LLM, our VLoRA can make LLM perceive visual information without extra computational overhead. To convert visual features into perceptual weights, we propose the perceptual weights generator to generate low-rank perceptual weights for any weights of LLM. Due to the low-rank property, the perceptual weights can be seen as LoRA weights, while $\Delta W_A$ is generated and $\Delta W_B$ is learnable. We perform comprehensive experiments on six MLLM benchmarks, and VLoRA can achieve comparable performance to LLaVA-v1.5 in most benchmarks while only bringing 10% computational cost as LLaVA's. In the ablation study, we reproduce LLaVA-v1.5 under the same settings and show that our method can achieve better performance.

# 7 Limitations

Despite VLoRA's promising performance on various benchmarks, it still has some limitations. 1) Representing images as model weights is a previously unexplored practice, and the extracted features from existing CLIP models may not be suitable to be converted into model weights. It is necessary to explore a vision encoder that is more suitable for this paradigm. 2) We use one perceptual weights generator for one type of weight, which may lead to an insufficient correlation between different types of generated perceptual weights. It may be better to use the same perceptual weights generator to produce weights for all types at once.

# Acknowledgments and Disclosure of Funding

This work was in part supported by the National Natural Science Foundation of China under grants 62032006 and 62021001, and by the Institute of Artificial Intelligence, Hefei Comprehensive National Science Center under grants 21KT013 and 23YGXT001. Mike Shou does not receive any funding for this work.

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

## A  Analysis of VLoRA computational overhead

In this subsection, we give a detailed calculation of the computational overhead of VLoRA. Similar to Sect. 3.4, we assume the LLM has $d$ blocks and hidden states dimension of $h$, the input text length is $C$, and the number of visual tokens is $L$. Therefore, the FLOPs of the self-attention module and the feed-forward network are $8Lh^2 + 4L^2h$ and $16Lh^2$. Since visual tokens are not introduced, then LLM has a computational overhead of $24Cdh^2 + 4C^2dh$ for text token sequence input. For training, we use perceptual weights as branches of LLM weights. The extra computation comes from three parts: 1) the matrix multiplication of the two perceptual weights with FLOPs of $24krh^2$. 2) The multiplication of the text token and the perceptual weights with FLOPs of $12Ckh^2$. 3) The output coming out of the perceptual weights is to be added to the output of the LLM weights with FLOPs of $14Ckh$. Therefore, the total FLOPs of VLoRA during training is $24Cdh^2 + 4C^2dh + 24krh^2 + 12Ckh^2 + 14Ckh$. For inference, we merge the perceptual weights with LLM's weights. The extra computation comes from two parts: 1) the matrix multiplication of the two perceptual weights with FLOPs of $24krh^2$, which is the same as training. 2) Adding perceptual weights to LLM weights with FLOPs of $12kh^2$. The total FLOPs during inference are $24Cdh^2 + 4C^2dh + 24krh^2 + 12kh^2$.

## B  Visualization Results

VLoRA can achieve promising results on various MLLM benchmarks, but these benchmarks are either multiple choice or judgmental, and to demonstrate VLoRA's capabilities even further, we show some real-world samples in Fig. 5. The first figure suggests that our VLoRA can count the accurate number of steaks in the image. The second figure shows that VLoRA has sufficient common sense. In the third figure, VLoRA demonstrates the ability to reason and have long text conversations.

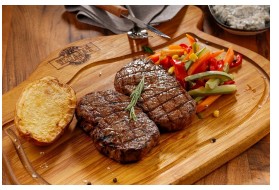
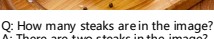

Q: How many steaks are in the image?
A: There are two steaks in the image?

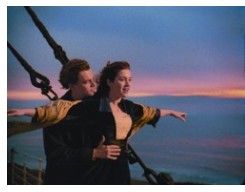

Q: Who is the director of this film?
A: James Cameron

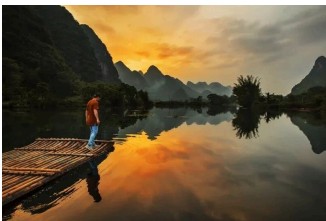

Q: Why is the river golden yellow?
A: The river appears golden yellow due to the reflection of the sunlight on the water. The sunlight creates a warm and vibrant glow on the surface of the river, which is then reflected onto the water's surface. This phenomenon is common in bodies of water, especially during sunrise or sunset, when the sunlight creates a beautiful and colorful atmosphere.

Figure 5: **Visualization results of VLoRA.** This figure demonstrates the capabilities of our VLoRA in real-world scenarios, including accurate counting and common sense reasoning.

## C  Broader Impacts

Our proposed new paradigm significantly improves the training and inference efficiency of multimodal large models and reduces the computational overhead, which, in terms of research, can reduce the resource threshold of multimodal large model research, which is conducive to the active exploration of researchers in related fields, and, in terms of practical application, reduces the cost of large-scale deployment for use and helps to reduce the consumption of resources.

## D  Comparisons on Fine-grained Benchmarks

We provide more results on fine-grained benchmarks for comprehensive comparisons, including TextVQA, DocVQA, InfoVQA, and OCRBench. As shown in Tab. 6, on these fine-grained benchmarks, VLoRA's performance has a gap compared to LLaVA on TextVQA and DocVQA, but it can achieve comparable results on InfoVQA. VLoRA converts CLIP's visual features into model weights, but CLIP's visual features are aligned with text rather than model parameters. Therefore, we need more diverse data to allow the weights generator to learn this transformation well. Since our pre-training data is coarse-grained image captioning data and the amount of fine-tuning data is limited, the performance of VLoRA trained on this dataset is not as good as LLaVA in some fine-grained tasks.

Table 6: Comparisons between VLoRA and LLaVA-v1.5 on fine-grained benchmarks, including TextVQA, DocVQA, InfoVQA, and OCRBench.

| Model | Size | # *vis. tok.* | TextVQA | DocVQA | InfoVQA | OCRBench | Avg. |
|---|---|---|---|---|---|---|---|
| LLaVA-v1.5 [34] | 7.2B | 576 | 58.2 | 18.4 | 20.4 | 31.8 | 28.0 |
| VLoRA | 7.8B | **0** | 51.4 | 13.4 | 19.5 | 27.7 | 25.8 |

Table 7: Comparisons of Training Speed and GPU Memory Requirements between VLoRA and LLaVA-v1.5

| | pre-training LLaVA | pre-training VLoRA | fine-tuning LLaVA | fine-tuning VLoRA |
|---|---|---|---|---|
| Training Speed | 106 samples/s | 246 samples/s | 46 samples/s | 73 samples/s |
| GPU RAM | 79G | 58.6G | 79G | 79G |

## E  Analysis of Training and Inference Efficiency

### E.1  Training Efficiency Analysis

As shown in Tab. 7, in the pre-training stage, the training speed of VLoRA can be 2.3 times faster than LLaVA. LLaVA's peak memory usage is 79G, while VLoRA's is significantly less at 58.6G. In the fine-tuning phase, VLoRA maintains a considerable advantage in training speed and can train 73 samples per second, 1.6 times faster than LLaVA. The memory usage of both is similar, around 79G, due to the learnable parameters of the LLM being the primary contributors to memory usage.

### E.2  Inference Efficiency Analysis

During the prefilling stage, VLoRA saves time by not calculating the kv cache for visual tokens. In the decoding stage, VLoRA decreases the time needed to calculate attention scores with visual tokens for each new token. As a result, VLoRA maintains an advantage in inference efficiency, even when generating long sentences.

**Prefilling stage.** Using a single A100 with flash attention, LLaVA takes 65 ms to produce the first token, whereas VLoRA only takes 45 ms. The primary time consumption for VLoRA is in weight generation, which has optimization potential, such as employing a single weight generator for all weight types.

**Decoding stage.** With a generated sequence length set at 256, and using Flash Attention, KV Cache, and Batch Inference to maximize speed on a single A100, the inference speed of LLaVA is 410 tokens per second. In contrast, VLoRA achieves 1078 tokens per second, which is 2.6 times faster than LLaVA.

