# OpenReview forum: "Visual Perception by Large Language Model’s Weights"
_NeurIPS.cc/2024/Conference — NeurIPS 2024 poster_

### Official Review · Reviewer_s5rD · 2024-07-08

**Soundness:** 2
**Presentation:** 3
**Contribution:** 2
**Rating:** 5
**Confidence:** 5

**Summary:**

This paper presents VloRA, a paradigm for building MLLMs, which aligns visual features with the parameter space of LLMs. By representing visual information as model weights, no visual tokens is need in the input, which reduces the length of the input sequence and improves efficiency.

**Strengths:**

(1) The motivation is convincing and the problem to solve is important. The enormous computational cost limits the training and inference devices for MLLMs.
(2) VLoRA significantly reduces the FLOPs of MLLMs both in training and inference. Moreover, although not mentioned in the paper, I believe VLoRA can also reduce the consumption of GPU RAM.
(3) Experimental results show that VLoRA maintains competitive performance.

**Weaknesses:**

(1) The paper presents the FLOPs advantage of VLoRA. However, FLOPs cannot reflect the real latency of LLMs sometimes, especially when generating a long sentence. The generation of LLMs has two stages: prefilling (calculating the KV cache and generating the first token) and decoding (generating subsequent tokens one-by-one). When generating a long sentence such as image captioning, shorter inputs can significantly reduce the prefilling time, but the decoding time is primarily determined by the length of the output.
(2)The experiments are not sufficiently comprehensive. To benchmark against LLaVA-1.5, the (zero-shot) comparisons should also be conducted on the following tasks: VQAv2, GQA, TextVQA, VisWiz, POPE, SEED, MM-Vet. I am particularly curious about the results of TextVQA because previous papers have shown that the performance on this dataset is strongly correlated with the number of visual tokens.
(3) VLoRA can be regarded as using vision features to generate the PEFT parameters (LoRA) of LLMs. Therefore, some similar work listed as following should be discussed in detail:

[1] HyperPELT: Unified Parameter-Efficient Language Model Tuning for Both Language and Vision-and-Language Tasks. ACL 2023 Findings.
[2] LLaMA-Adapter: Efficient Fine-tuning of Language Models with Zero-init Attention. ICLR 2024.
[3] Memory-Space Visual Prompting for Efficient Vision-Language Fine-Tuning. ICML 2024.

**Questions:**

(1) Please provide the experimental data about the real training speed and GPU RAM requirement of VLoRA and LLaVA.
(2) Please provide the experimental data on the inference efficiency of VLoRA and LLaVA, especially on long-sequence generation.
(3) Some recently similar work should be discussed in detail.
(4) How about the results on VQAv2, GQA, TextVQA, VisWiz, POPE, SEED, MM-Vet.
(5) Why use CapsFus-30m instead of blip-558k for pretraining? Can VeLoRA still be competitive when using a smaller blip-558k for pretraining?

**Limitations:**

This paper discusses some of the method's limitations, but I believe the discussion can be more comprehensive. See the 'Weaknesses' mentioned above.

---

> ### Author Rebuttal · Authors · 2024-08-07
>
> We are deeply grateful for the reviewer's valuable comments.
>
> **Question 1: The experimental data about the real training speed and GPU RAM requirement of VLoRA and LLaVA**
>
> |                | pre-training LLaVA | pre-training VLoRA | fine-tuning LLaVA | fine-tuning VLoRA |
> | - | - | - | - | - |
> | training speed | 106 samples/s | 246 samples/s | 46 samples/s | 73 samples/s |
> | GPU RAM        | 79G              | 58.6G            | 79G             | 79G             |
>
> 1) In the pre-training stage, the training speed of VLoRA can be 2.3 times faster than LLaVA. And LLaVA's peak memory usage is 79G, while VLoRA's is significantly less at 58.6G.
>
> 2. In the fine-tuning phase, VLoRA still maintains a considerable advantage in training speed and can train 73 samples per second, which is 1.6 times faster than LLaVA. The memory usage of both is similar, around 79G, due to the learnable parameters of the LLM being the primary contributors to memory usage.
>
> **Question 2:  The experimental data on the inference efficiency of VLoRA and LLaVA, including long-sequence generation.**
>
> During the prefilling stage, VLoRA saves the time of calculating the kv cache of visual tokens. In the decoding stage, VLoRA decreases the time needed to calculate attention scores with visual tokens for each new token. Therefore, even when generating a long sentence, VLoRA's inference efficiency still has an advantage.
>
> We compare the inference speed of VLoRA and LLaVA on a single A100, and utilize KV Cache, FlashAttention, and Batch Inference techniques to achieve the maximum practical inference speed.
>
> | num of generated tokens | VLoRA (tokens/s) | LLaVA (tokens/s) | Speed Ratio |
> | - | - | - | - |
> | 256                     | 1078             | 410              | 2.6         |
> | 512                     | 865              | 342              | 2.5         |
> | 1024                    | 451              | 250              | 1.8         |
>
> With a generated sequence length of 256, VLoRA's generation speed is 1078 tokens/s, 2.6 times faster than LLaVA. At a length of 512, VLoRA remains 2.5 times faster. We find that VLoRA still maintains an advantage as sequence length increases. Even at a length of 1024, VLoRA's speed is 1.8 times faster than that of LLaVA.
>
> **Question 3: Discussion with recent similar work.**
>
> We greatly appreciate the relevant work that the reviewer has highlighted. We will integrate these discussions into our paper. The following are our discussions.
>
> LLaMA-Adapter inserts learnable prompts into L of N decoder layers and uses zero-initialized attention for stable training. HyperPELT employs a shared hypernetwork that generates weights for fine-tuning various modules. MemVP concatenates visual prompts with FFN weights for injecting visual knowledge. In contrast, 1) VLoRA can inject visual information at any linear module, offering flexibility. 2) Unlike task-level PEFT methods, VLoRA is sample-level, generating weights for individual input images. Our evaluations, mainly in zero-shot settings, demonstrate VLoRA's strong generalization ability.
>
> **Question 4: The zero-shot comparisons results on VQAv2, GQA, TextVQA, VisWiz, POPE, SEED, MM-Vet.**
>
> These datasets are more fine-grained, but among them, VQAv2, GQA, and TextVQA are not zero-shot. To make a  zero-shot comparison, we also evaluated on other zero-shot fine-grained datasets, including OCRBench, AI2D, InfoVQA, MathVision (math-related), SeedBench-2 (SeedBench series), SeedBench-2 Plus (Text-related), and BLINK (difficult visual perception tasks).
>
> | Method | textvqa | docvqa | vqav2 | gqa  | viswiz | pope | seed | mmvet | ocrb | ai2d | infovqa | mathvision | seed2 | seed2 plus | blink | Avg. |
> | - | - |  - | - | - | - | - | - | - | - | - | - | - | - | - | - |  - |
> | LLaVA  | 58.2 | 18.39 | 78.5 | 61 | 50 | 86.1 | 65.8 | 32.9 | 31.8 | 55.5 | 20.4 | 8.52 | 43.4 | 40.05 | 39.7 |46.1|
> | VLoRA  | 51.43 | 13.41 | 71.5 | 51.42 | 41.31 | 80.5 | 54.6 | 25.8 | 27.7 | 54.01 | 19.46 | 11.7 | 44.92 | 37.5 | 39.8 | 41.7 |
>
> We find that VLoRA's performance on TextVQA, DocVQA, and VQAv2 has a gap compared to LLaVA, but on other fine-grained benchmarks such as AI2D and InfoVQA, its performance is comparable to LLaVA's.
>
> Therefore, although VLoRA has high training and inference efficiency and achieves considerable performance on general benchmarks, there is still room for improvement in this method on fine-grained benchmarks.
>
> The possible reasons are: 1) The lack of diverse training data. VLoRA transforms CLIP's visual features into model weights, but CLIP's visual features are aligned with text rather than model weights. Therefore, diverse data is needed to allow the weights generator to retain sufficient visual information when transforming visual features into model weights. However, VLoRA is pre-trained on CapsFus-30M, a coarse-grained image captioning data, which limited the performance of VLoRA.  2) The data ratio has not been adjusted. The ratio of different types of data is crucial to the performance of MLLMs. Our model architecture is completely different from methods like LLaVA that align visual tokens to the input space, so the data ratio should be readjusted.
>
> The purpose of VLoRA is to provide an efficient new parameter-space-aligned MLLM paradigm, and in this paper, we focus on general scenarios. Compared to the well-developed LLaVA, there are still many areas to explore in this new paradigm, including training data and visual encoders, which are also part of our future work.
>
> **Question 5: The reason of using CapsFus-30M instead of blip-558k for pretraining.**
>
> The reason is that our weights generator has more learnable parameters than LLaVA's projector, and needs to learn to transform CLIP's visual features into model weights. Therefore, we need to use CapsFus-30M instead of blip-558k. If only pre-trained with blip-558k, the generator struggles to convert visual features into model weights, resulting in reduced VLoRA performance.

---

> ### Comment · Area_Chair_32Ep · 2024-08-12
> **Concerns addressed?**
>
> Dear reviewer, thank you for a thoughtful review! Are your concerns about relevant metrics and evaluation on tasks where performance strongly correlates with number of visual tokens addressed in the rebuttal?

---

### Official Review · Reviewer_MX94 · 2024-07-12

**Soundness:** 3
**Presentation:** 3
**Contribution:** 3
**Rating:** 6
**Confidence:** 5

**Summary:**

The paper proposes a novel way to enable visual understanding in LLMs. Instead of encoding image as visual tokens, the paper proposes converting visual input to low-rank perceptual weights which are merged with LLM weights (similar to LoRA). The paper shows that the proposed approach achieves comparable performance on various single image V+L benchmarks, while significantly reducing computational cost.

**Strengths:**

The paper presents a novel idea of encoding visual information as low-rank perceptual weights instead of visual tokens. It's a fresh perspective on how to integrate visual knowledge in LLMs which hasn't been done before.
- The results show that the performance is on-par with existing methods on most standard V+L benchmarks, while requiring significantly less computational overhead (as measured by GFLOPS).
- The authors show exhaustive ablations for the perceptual weights generator which were quite insightful.

**Weaknesses:**

While the method is interesting, and novel,  several practical questions remain that affect the flexibility of that method:
- How will the model work when more than one image is used as input (such as interleaved image-text dialogue, videos, etc).
- While the method achieves better GFLOPS than existing paradigm of using visual tokens, practical advancements (FlashAttention, KVCaching) significantly reduce actual computational overhead of adding more tokens. Can the authors comment on how their model compares after accounting for these tricks that people use to speed up inference? The authors can consider reporting tokens/s, and time to first token instead of GFLOPs.
- I would have liked to see results on benchmarks which require fine-grained image understanding and spatial understanding and will potentially benefit from high-resolution like TextVQA, OKVQA, DocVQA.

**Questions:**

- Can the authors comment on memory overhead during training? I imagine that storing the weights of perceptual weights generator (especially using different ones for different weight type) is expensive?
- What is the red dotted vertical line in Figure 4 (Left)

**Limitations:**

The authors discuss some limitations, but miss many as pointed out in the weakness section. E.g., theoretical vs practical benefit when utilising modern tricks to speed up inference, as well as how to model multiple images.

---

> ### Author Rebuttal · Authors · 2024-08-07
>
> We are grateful for the effort the reviewer has dedicated to evaluating this work.
>
> **Question 1: How will the model work when more than one image is used as input (such as interleaved image-text dialogue, videos, etc).**
>
> VLoRA can naturally be extended to support multiple image inputs, here we consider three scenarios for multi-image input.
>
> **1) Multimodal In-Context Learning.** In this scenario, the input to MLLM will provide multiple image-text pairs as examples to assist in answering the query image and question. We can use the weights generator to create multimodal LoRA weights for each in-context example, using both image and text as input. For the query image and question input, we generate query LoRA weights  from the query image, and the question is input to the LLM.
>
> **2) Interleaved Image-Text Dialogue.** In this case, images and text have a temporal relationship. Given input $C = \{V_1, T_1,..., V_N, T_N\}$, we generate $N$ LoRA weight sets, $W = \{W_1,..., W_N\}$, corresponding to $N$ images. Text $T=\{T_1, T_2, ..., T_N\}$ is input to the LLM. During training and inference, tokens of $T_j$ pass through the matrix of $W_i$ where $i ≤ j$. For instance, $T_1$ tokens pass through $W_1$ and $T_2$ tokens pass through $W_1$ and $W_2$. This ensures text tokens only attend to preceding image information, maintaining causality.
>
> **3) Video Input.** We can represent different video frames with multiple sets of LoRA weights, and in order to maintain the temporal relationship between video frames, we can add learnable position encodings to the different LoRA weights. We can also extract the representation of the entire video, and then generate a single LoRA weight from the video representation to represent the video information.
>
> VLoRA offers a new MLLM paradigm. This paper focuses more on validating its feasibility in general scenarios, and the extension to multi-image input can be considered as future work for further exploration.
>
> **Question 2: Compare VLoRA's efficiency considering practical advancements like FlashAttention and KVCaching, and report tokens per second and time to first token.**
>
> When using the same acceleration techniques, compared to LLaVA, VLoRA still has significant efficiency advantages during both training and inference. We discuss the efficiency of VLoRA during the training and inference, and compare with LLaVA under the same machine.
>
> **Training efficiency.** The training has pre-training and fine-tuning stages, and Flash Attention technique was used for training.
>
> |                | pre-training LLaVA | pre-training VLoRA | fine-tuning LLaVA | fine-tuning VLoRA |
> | - | - | - | - | - |
> | training speed | 106 samples/s     | 246 samples/s | 46 samples/s | 73 samples/s |
>
> In the pre-training stage, VLoRA can be 2.3 times faster than LLaVA. In the fine-tuning stage, VLoRA still maintains a considerable advantage and can train 73 samples per second, which is 1.6 times faster than LLaVA.
>
> **Inference efficiency.** In the prefilling stage, VLoRA reduces the time of calculating the kv cache of visual tokens. In the decoding stage, VLoRA reduces the time to compute attention scores with visual tokens for each new token.
>
> 1) Prefilling stage. Using a single A100 with flash attention, LLaVA's time to first token is **65 ms**, while VLoRA's is **45 ms**. VLoRA's primary time consumption is in weight generation, which has optimization potential, such as using a single weights generator for all weights type.
>
> 2. Decoding stage. We set the generated sequence length at 256 and employ Flash Attention, KV Cache, and Batch Inference to achieve the maximum inference speed. The inference is performed on a single A100. The inference speed of LLaVA is **410 tokens/s**, while that of VLoRA is **1078 tokens/s**, which is **2.6 times** that of LLaVA.
>
> **Question 3: More results on fine-grained benchmarks.**
>
> We provide more results on fine-grained benchmarks, including TextVQA, DocVQA, and other fine-grained benchmarks, like OCRBench and InfoVQA. Due to time constraints, we can't complete the evaluation of OK-VQA  in time to provide results.
>
> | method | OCRBench | InfoVQA | TextVQA | DocVQA | Average |
> | - | - | - | - | - | - |
> | LLaVA  | 31.8     | 20.4    | 58.2    | 18.4   | 28.0    |
> | VLoRA  | 27.7     | 19.5    | 51.4    | 13.4   | 25.8    |
>
> On these fine-grained benchmarks, VLoRA's performance has a gap compared to LLaVA on TextVQA and DocVQA, but it can achieve comparable results on InfoVQA. VLoRA converts CLIP's visual features into model weights, but CLIP's visual features are aligned with text rather than model parameters. Therefore, we need more diverse data to allow the weights generator to learn this transformation well. Since our pre-training data is coarse-grained image captioning data and amount of fine-tuning data is limited, the performance of VLoRA trained on this dataset is not as good as LLaVA in some fine-grained tasks.
>
> The purpose of VLoRA is to provide a new parameter-space-aligned MLLM paradigm, and we focus more on the general scenarios. Compared to the well-developed LLaVA, there are still many areas to explore in this new paradigm, including training data and visual encoders, which are also part of our future work.
>
> **Question 4: Memory overhead during training and inference.**
>
> VLoRA's memory usage is lower compared to LLaVA. LLaVA's 576 visual tokens per layer result in higher memory overhead than VLoRA's weight generators.  In pre-training, LLaVA uses 79G, and VLoRA uses 58.6G memory. During fine-tuning, both use approximately 79G due to LLM's learnable parameters. In inference, with a batch size of 16 and sequence length of 512, LLaVA uses 39G, and VLoRA uses 35G memory.
>
> **Question 5: The meaning of the red dotted vertical line in Figure 4 (left)**
>
> The red dotted vertical line in Figure 4 (left) represents the position where the number of visual tokens is 576, which is the number of input visual tokens for LLaVA.

---

> ### Comment · Area_Chair_32Ep · 2024-08-12
> **Concerns addressed?**
>
> Dear reviewer, thank you for a well thought out review! Are you concerns above extension to multiple images, gains in context of FlashAttention/KVCache, and fine grained image understanding satisfactorily addressed?

---

> > ### Comment · Reviewer_MX94 · 2024-08-12
> > **Thanks for the rebuttal.**
> >
> > Thank you for answering all my questions. New analysis to measure training and inference efficiency (requested by multiple reviewers) is much appreciated and makes the paper more thorough. It was also great to see experiments on more fine-grained benchmarks. Even though the numbers are lower than LLAVA, it shows opportunities for future work. I have increased the score to 6 (Weak Accept).

---

### Official Review · Reviewer_cXpd · 2024-07-13

**Soundness:** 4
**Presentation:** 4
**Contribution:** 2
**Rating:** 5
**Confidence:** 4

**Summary:**

The work proposes an efficient setup for incorporating non-text modalities into pretrained LLMs for reasoning-based tasks. Instead of introducing new tokens into the LLM, they propose to dynamically generate LoRA weight matrix residuals for the linear projectors within the LLM, conditioned on the input image. The weight matrix updates, when applied to the LLM, will then alter how the LLM processes the input text.

Experimental results demonstrate competitive performance on QA tasks without the quadratic cost associated with increased input tokens.

**Strengths:**

- Simple and efficient approach with effective results
- Detailed analysis on architecture and the effect of the rank of the update matrix.

**Weaknesses:**

- I think it's worthwhile to look beyond the QA benchmark numbers to understand what the implications of this architectural change are. If I am to understand this method correctly, the weight generator is not conditioned on the text-to-be-ingested by the LLM, which means it could drop information not typically useful for the task it's trained on. It would be curious to see how this approach compares to the more standard approach when it comes to asking questions about very obscure (or spatially tin) elements within an image.
- The approach also appears to have a pretty sensitive sweet spot for rank, which could be expensive to tune for
- From the perspective of technical novelty, I believe this is closely related to HyperNetworks https://arxiv.org/pdf/1609.09106 of which there are also transformer variants: https://arxiv.org/pdf/2106.04489 . The authors should probably include a relevant related-works section for this as well, and perhaps some additional comparisons against adaptation techniques proposed there as well. I think the goal would be to demonstrate that the proposed approach works best when it comes to sample-level adaptations, as compared to the typical task-level adaptations.

**Questions:**

See weaknesses

**Limitations:**

Yes

---

> ### Author Rebuttal · Authors · 2024-08-07
>
> We appreciate the insightful comments provided by the reviewer.
>
> **Question 1: Look beyond QA benchmark numbers to understand what the implications of this architectural change are.**
>
> Thank you for your suggestion. We have provided some practical examples in the PDF file, where you can see the impact of architectural changes. We conducted tests on practical examples. From the samples on the left, we can see that both VLoRA and LLaVA can recognize the fine-grained car logos in the image. However, for the text recognition on the right, both models ignored the target area in the instruction and made mistakes in their answers.
>
> **Question 2: Whether weights generator could drop information not typically useful for the task it's trained on.**
>
> Our weights generator of VLoRA is conditioned solely on the input image, which requires the generated LoRA weights to contain as much comprehensive image information as possible, rather than pre-extracting the information needed for LLM based on the input text. This design is beneficial for the model's generalization ability.
>
> However, weights generator also requires a large amount of diverse pre-training data to train the model. In situations where the diversity and quantity of data are insufficient, the weights generator may lose necessary information. Our model is pre-trained only on the image captioning dataset, where diversity is not guaranteed, so it is possible to lose some information. To measure this potential loss, we evaluated our model on some zero-shot fine-grained or unconventional benchmarks. We evaluated our model on Text-central benchmarks like AI2D, InfoVQA and Seed2 Plus, and mathematical benchmark MathVision (math-related tasks are rare in training data), BLINK (tasks like multi-perspective reasoning, depth estimation, and reflexive estimation, which are also rare in training data).
>
> | method | AI2D  | InfoVQA | Seed2 Plus | MathVision | Blink | Average |
> | ------ | ----- | ------- | ---------- | ---------- | ----- | ------- |
> | LLaVA  | 55.5  | 20.4    | 40.05      | 8.52       | 39.7  | 32.8    |
> | VLoRA  | 54.01 | 19.46   | 37.5       | 11.7       | 39.8  | 32.5    |
>
> The results show that VLoRA's performance on unconventional tasks is comparable, but there is still a gap in text-related tasks. The reason is our pre-training data is limited and only consisting of coarse-grained image captioning data.
>
> **Question 3: It would be curious to see how this approach compares to the more standard approach when it comes to asking questions about very obscure (or spatially tin) elements within an image.**
>
> We provided serveal examples in Q1, and in Q2, we perform results on zero-shot fine-grained or unconventional benchmarks.
>
> **Question 4: Whether this approach is sensitive to the setting of rank, which could be expensive to tune for.**
>
> Our approach is not sensitive to the setting of rank.  In Tab 4, for a fair comparison, we use the same dataset Capsfusion-30M for pre-training. This dataset is insufficient for the scenario where the rank is set to 128, leading to a performance degradation in this case. A higher rank signifies that the perceptual weights generated by the weights generator have a larger dimension, which poses a greater demand on the weights generator. Consequently, a larger amount of pre-training data is required to effectively train the weights generator. In the below table,  we increase the pre-training data to 60M, it can be observed that the performance of a rank of 128 becomes comparable to that of a rank of 64.
>
> | rank | data           | MMBench | MME          | ScienceQA | HallusionBench | MMMU | CCBench | Average |
> | ---- | -------------- | ------- | ------------ | --------- | -------------- | ---- | ------- | ------- |
> | 64   | Capsfusion-30M | 63.4    | 1311.3(65.6) | 66.4      | 26.4           | 36.0 | 28.6    | 47.7    |
> | 128  | Capsfusion-30M | 61.0    | 1228.4(61.4) | 68.0      | 23.8           | 33.4 | 26.7    | 45.7    |
> | 128  | Capsfusion-60M | 62.8    | 1337(66.9)   | 65.5      | 25.8           | 33.4 | 30.2    | 47.4    |
>
> **Question 5: Discussion with more related work.**
>
> We very appreciate the related work reviewer have pointed out. We will incorporate these discussions into our paper. Below are our discussions.
>
> HyperNetworks proposes static hypternetwork for CNN and dynamic hypternetwork for RNN. HyperFormer proposes hypterformer to generate adapter parameters for all layers and multiple tasks using shared hypternetworks.  The parameter generation of both methods is designed on task-level for pre-defined tasks.
>
> Different from them,  1) VLoRA focuses on sample-level parameter generation, the generated LoRA weights are conditioned on the input image without pre-defining tasks during training. Because the target of MLLM is to address a variety of  tasks or problems, which are difficult to fully define in advance. Therefore, task-level adaption is unsuitable for recent MLLM. 2) VLoRA utilizes the generated parameters in LoRA way. Sample-level parameter generation can lead to significant changes in model parameters. VLoRA, adopting the LoRA method, can better maintain the inherent capability of the pre-trained LLM.

---

> ### Comment · Area_Chair_32Ep · 2024-08-12
> **Concerns addressed?**
>
> Dear reviewer, thank you for a thoughtful review! are your concerns addressed by the rebuttal?

---

> > ### Comment · Reviewer_cXpd · 2024-08-12
> > **No further questions**
> >
> > Thank you, I believe my concerns have been adequately addressed and I will take into account the additional information in the final discussion phase with other reviewers.

---

### Author Rebuttal · Authors · 2024-08-07

The figure requested by reviewer cXpd has been included in the PDF

---

### Decision · Program_Chairs · 2024-09-25

**Decision:**

Accept (poster)

**Comment:**

The work presents a novel approach for integrating visual inputs in an LLM by predicting and fusing weights instead of features. The work initially received 3BA. Post-rebuttal one of the reviewers raised their rating to WA. In the reviewer discussion phase, the reviewers showed their support for acceptance in spite of some limitations (not all results being SOTA and other practical considerations). The reviewers especially appreciated the "fresh perspective with substantial experiments and analysis", "good insights",  and the work "testing out a rather simple idea, as opposed to many recent papers which present larger, complex, and often ad-hoc solutions with very little insight".

Having read the reviews, rebuttal and the following discussions, the AC agrees with the reviewers and recommends "accept". Congratulations to the authors!